# A universal approach to Krylov state and operator complexities

**Mohsen Alishahiha[1] and Souvik Banerjee[2,3]**

**1** School of Physics, Institute for Research in Fundamental Sciences (IPM),
P.O. Box 19395-5531, Tehran, Iran
**2** Institut für Theoretische Physik und Astrophysik, Julius-Maximilians-Universität Würzburg,
Am Hubland, 97074 Würzburg, Germany
**3** Würzburg-Dresden Cluster of Excellence ct.qmat

## Abstract

We present a general framework in which both Krylov state and operator complexities can be put on the same footing. In our formalism, the Krylov complexity is defined in terms of the density matrix of the associated state which, for the operator complexity, lives on a doubled Hilbert space obtained through the channel-state map. This unified definition of complexity in terms of the density matrices enables us to extend the notion of Krylov complexity, to subregion or mixed state complexities and also naturally to the Krylov mutual complexity. We show that this framework also encompasses nicely the holographic notions of complexity.



## 1 Introduction

Quantum mechanics provides us with two apparently different yet natural ways to understand the complexity of a system through time-evolution. In the Schrödinger picture which allows

the state to be time-dependent, complexity measures the mixing of the initial state with other states through the time-evolution, which for a time independent Hamiltonian $H$ takes the form

$$|\psi(t)\rangle = e^{iHt}|\psi(0)\rangle = \sum_{n=0}^{\infty} \frac{(it)^n}{n!}|\psi^{(n)}\rangle, \tag{1}$$

where $|\psi^{(n)}\rangle \equiv H^n|\psi(0)\rangle$. Thus, in this notation, complexity of the final state amounts to understanding the spread of the corresponding wavefunction in a fixed basis. The choice of an optimal basis here is a bit tricky. One interesting way proposed in [1] where the minimization of the spreading of the wavefunction determines the optimal basis. This notion of state complexity was coined as the "spread complexity".

Alternatively, one may wish to switch to the Heisenberg picture where the time evolution is attributed to the operator instead. This is realised by noting the expectation value of an operator in the complexified state,

$$\langle\psi(t)|\mathcal{O}|\psi(t)\rangle = \langle\psi(0)|e^{-iHt}\mathcal{O}e^{iHt}|\psi(0)\rangle, \tag{2}$$

and immediately recognizing the time dependent operator

$$\mathcal{O}(t) = e^{-iHt}\mathcal{O}e^{iHt} = \sum_{n=0}^{\infty} \frac{(it)^n}{n!}\mathcal{O}^{(n)}, \tag{3}$$

where $\mathcal{O}^{(n)}$ are understood as nested structures of operators, $\mathcal{O}^{(n)} \equiv [H, \cdots, [H, \mathcal{O}]\cdots]$ and provide the notion of operator complexity by interpreting $\mathcal{O}(t)$ as the operator wave function evolved by a Liouvillian superoperator $\mathcal{L}$

$$\mathcal{O}(t) = e^{i\mathcal{L}t}\mathcal{O}, \tag{4}$$

with $\mathcal{L} = [H, \cdot]$. In this notation, $\mathcal{O}^{(n)} \equiv \mathcal{L}^n\mathcal{O}$ which determines the mixing of operators. In order to compute complexity corresponding to the growth of the operator, one uses the Lanczos algorithm [2] to construct an optimal basis [3], known in literature as the Krylov basis. The corresponding operator complexity is termed as the Krylov complexity [4].

Last but not the least, in the present decade there has been one more entry in the world of complexity in form of the holographic complexity. As its name suggests, this notion of complexity arose out of the curiosity to understand the interior of a black hole spacetime in the light of AdS/CFT correspondence [5–7]. In particular, the fact that the volume the interior of the black hole keeps growing even after attaining thermal equilibrium [8,9] is very much reminiscent of the nature of complexity of a finite entropic fast-scrambling system.

Motivated by this striking similarity, a holographic definition of complexity was proposed as the volume of the maximal slice in the interior of the black hole. This proposal is celebrated in the name "Complexity = Volume" (CV) conjecture [8,10]. An efficient formalism to study this interior volume in two dimensional theory of gravity was developed in [11–13] which produces the expected behaviour of the late-time linear growth and eventual saturation of complexity.

This observation was formalized in a more general context in [14], which, based on the Eigenstate Thermalization Hypothesis (ETH) [15,16], classified the possible candidates for complexity which exhibit a linear growth at late time. It was also shown that this class of observables naturally includes the expectation value of the quenched length operator defined in [11–13].

It is a daunting task to bring all these apparently different notions of complexity under the same umbrella. And this is precisely the motivation of the present work.

In the process of doing so, we will first present, **in section 2**, a general framework to study complexity of a given state. This state can either be a given autonomous state or could as well be a state evolved under the evolution by a general Hermitian operator which is not necessarily the Hamiltonian of the system. In either case, for a given orthonormal and ordered basis, we can define a label operator and subsequently a number obtained by tracing the former over the density matrix corresponding to the given or the evolved state. This number is minimized over the space of operators used to generate sets of the orthonormal basis through Gram-Schmidt process, to define the complexity of the state.

These operators can be identified as different quantum gates in accordance with the quantum informatic definition of circuit complexity [17]. When the state is created through a unitary evolution, this definition of complexity naturally yields the Krylov complexity.

We show that, for the Hamiltonian evolution, the class of observables identified in [14], with the desired pole structure to produce the late-time linear growth of complexity, can be identified, naturally, with the expectation value of the label operator mentioned above. We further show that our formalism also naturally provides the late time saturation of complexity following the linear growth.

**In section 3**, we develop the formalism to study operator complexity. The structure of the label operator dictates a mapping of the space of operators to a doubled Hilbert space endowed with a inner product structure. This enables us to recast the operator complexity associated with any given operator to the complexity of a state in the doubled Hilbert space. This definition is also consistent with the Liouvillian evolution for operator complexity.

From the perspective of quantum information, the mapping to the doubled Hilbert space is a realisation of the channel-state map. However, we demonstrate that it has a beautiful interpretation in axiomatic quantum field theory and reveals a deeper structure of entanglement in the Hilbert space which turns out to be pivotal in understanding the holographic notion of complexity.

**In section 4** we discuss a novel biproduct of our generalized formalism for studying the state and the operator complexities. Since we defined complexity in terms of density matrices, it fits in as the ideal candidate to study subregion complexity using reduced density matrix of a given subregion. Definition of subregion Krylov complexity reveals some more interesting and fundamental aspects connecting quantum entanglement and the growth of complexity.

Finally, we conclude **in section 5** with some interesting open questions and some works in progress.

## 2 Complexity: The general framework

Be it for a state or for an operator, the general strategy to compute complexity comprises of the following steps - i) start with an initial state (operator), ii) allow it to spread over some state (operator) basis via an evolution generated by a time-independent Hermitian operator, iii) define a quantity that could probe the spreading while iv) the most efficient spreading is quantified by the minimization of the afore-mentioned quantity which is equivalent to finding a basis for the state (operator) so that the spreading becomes minimum.

In this section we will develop this general framework for state complexity. In the section to follow, we will extend this algorithm to study operator complexity.

**Complexity of a generic given state:** Let us consider a quantum system described by a time independent Hamiltonian whose eigenstates and eigenvalues are denoted by $|E_a\rangle$ and $E_a$, respectively. Here $a = 1, 2, \cdots \mathcal{D}$ with $\mathcal{D}$ being the dimension of the associated Hilbert space $\mathcal{H}$.

For a given Hermitian operator, $\mathcal{A} : \mathcal{H} \to \mathcal{H}$, one can construct an orthonormal and ordered basis associated with any state of this Hilbert space. Denoting the corresponding state by $|\psi\rangle$, the ordered orthonormal basis $\{|n\rangle, n = 0, 1, 2, \cdots, \mathcal{D}_\psi - 1\}$ can be constructed using the Gram-Schmidt process. The first element of the basis is the given state of the Hilbert state $|0\rangle = |\psi\rangle$ which we assume to be normalized. Then the other elements are constructed recursively as follows

$$\widehat{|n+1\rangle} = (\mathcal{A} - a_n)|n\rangle - b_n|n-1\rangle\,, \tag{5}$$

where $|n\rangle = b_n^{-1}|\hat{n}\rangle$ and

$$a_n = \langle n|\mathcal{A}|n\rangle\,, \qquad b_n = \sqrt{\langle\hat{n}|\hat{n}\rangle}\,. \tag{6}$$

This recursive procedure stops whenever $b_n$ vanishes which occurs for $n = \mathcal{D}_\psi$ defined as the dimension of subspace $\mathcal{H}_\psi$ expanded by the basis $\{|n\rangle\}$. The dimension of $\mathcal{H}_\psi$ is in general smaller than the dimension of the full Hilbert space: $\mathcal{D}_\psi < \mathcal{D}$. Note that this procedure produces an orthogonal basis together with coefficients $a_n$ and $b_n$ known as the Lanczos coefficients [2].

Since the basis of the subspace $\mathcal{H}_\psi$ is an ordered basis by construction, one can label any element of the subspace by a number which amounts to defining a label operator as

$$\ell = \sum_{n=0}^{\mathcal{D}_\psi - 1} c_n|n\rangle\langle n|\,, \tag{7}$$

for arbitrary functions $c_n$ which is the "label" associated with the state $|n\rangle$. Note that for $n > n'$ one assumes $c_n > c_{n'}$. Since the basis $\{|n\rangle\}$ is already ordered, a natural choice for the coefficient $c_n$ is $c_n = n$.

By construction, the basis $\{|n\rangle\}$ defines a complete basis for the subspace $\mathcal{H}_\psi$. Therefore, any state $|\phi\rangle \in \mathcal{H}_\psi$ can be expanded as

$$|\phi\rangle = \sum_{n=0}^{\mathcal{D}_\psi - 1} \phi_n|n\rangle\,, \quad \text{with} \quad \sum_{n=0}^{\mathcal{D}_\psi - 1} |\phi_n|^2 = 1\,. \tag{8}$$

The expectation value of the label operator in this state $|\phi\rangle$ is given by

$$\langle\phi|\ell|\phi\rangle = \mathrm{Tr}(\ell\rho_\phi) = \sum_{n=0}^{\mathcal{D}_\psi - 1} n|\phi_n|^2\,, \tag{9}$$

where $\rho_\phi = |\phi\rangle\langle\phi|$ is the density matrix associated with the state $|\phi\rangle$.

Using this expression, one can assign a "spreading number" to a given state $|\phi\rangle$ in terms of its density matrix as

$$\mathcal{C}_\phi = \mathrm{Tr}(\ell\rho_\phi)\,. \tag{10}$$

As its name suggests, (10) can be thought of as a quantity that measures the spreading of state $|\phi\rangle$ in the orthogonal basis $\{|n\rangle\}$. Note that, in this notation, the spreading of the state $|m\rangle \in \{|n\rangle\}$ is $m$. Since the above definition of spreading number is given in terms of the density matrix, it can be naturally extended for mixed states as well. We will come back to this point later in this paper.

For large $\mathcal{D}_\psi$ one would expect that for a typical state given by (8) with maximum spreading, the spreading is distributed statistically over all elements of the basis with equal probability: $|\phi_n|^2 \sim \frac{1}{\mathcal{D}_\psi}$ [18]. From this argument, the maximum value for spreading of a state can be estimated as

$$\mathrm{Tr}(\ell\rho_\phi) \sim \frac{\mathcal{D}_\psi}{2}\,. \tag{11}$$

Although already evident from our construction, it is worth emphasizing that the spreading number we have associated with the state $|\phi\rangle \in \mathcal{H}_\psi \subset \mathcal{H}$ depends on two ingredients: the original state $|\psi\rangle$ and the operator $\mathcal{A}$ by which the ordered basis is constructed. This is very reminiscent of the computational complexity in quantum information theory [17] in the sense that the original state $|\psi\rangle$ plays the role of the reference state while operator $\mathcal{A}$, or equivalently the ordered basis $\{|n\rangle\}$, can be thought of as quantum gates.

Endowed with this interesting identification, we can go ahead to define complexity using (10) as follows. After fixing the reference state, $|\psi\rangle$, one can construct the ordered basis using different Hermitian operators which can be thought of as considering different gates. If one can find a Hermitian operator among all possible operators, the basis constructed from which minimizes the spreading number (10), then the corresponding spreading number we will define as the complexity of the state. Therefore, in this context, finding complexity of a state boils down to the problem of finding the optimal operator, $\mathcal{A}_{\text{opt}}$.

We note, however, that in general, for given reference and target states, this minimization procedure to find $\mathcal{A}_{\text{opt}}$ is a complicated program. However, we will show now that for particular cases in which the state is obtained by a unitary transformation from the reference state, this can be successfully achieved.

**Complexity following a unitary evolution:** So far we considered a typical autonomous state $|\phi\rangle$ without knowing apriori whether this state was obtained through any dynamical process from an initial reference state. We will now focus on the case where the desired (target) state is obtained from a reference state via a unitary transformation. In this case it is straightforward to generalize the notion of spreading number as follows.

Let us consider the following state

$$|\phi\rangle \equiv |\psi(s)\rangle = U(\tilde{\mathcal{A}}, s)|\psi\rangle, \tag{12}$$

whose evolution in the parameter space $s$ is governed by the Schrödinger-like equation

$$i\frac{d}{ds}|\psi(s)\rangle = \mathcal{U}(\tilde{\mathcal{A}}, s)|\psi(s)\rangle, \tag{13}$$

with $\mathcal{U}(\tilde{\mathcal{A}}, s) = i\frac{d}{ds}U(\tilde{\mathcal{A}}, s)U^{-1}(\tilde{\mathcal{A}}, s)$. Here the unitary operator $U$ which evolves the state from an initial reference state $|\psi\rangle \equiv |\psi(0)\rangle$, is a function of the Hermitian operator $\tilde{\mathcal{A}}$ and the parameter $s$ which defines the flow through (13). The latter can be chosen such that at $s = 0$ one has $U = 1$. Following its definition given in (10), the spreading number at any arbitrary point $s$ on the flow is then given by

$$\mathcal{C}(s) = \text{Tr}(\ell\rho(s)) = \sum_{n=0}^{\mathcal{D}_\psi - 1} n \langle n|\psi(s)\rangle\langle\psi(s)|n\rangle, \tag{14}$$

where $\rho(s) = |\psi(s)\rangle\langle\psi(s)|$ is the density of state at $s$ and the label operator $\ell$ is constructed using a set of orthonormal basis $\{|n\rangle\}$ which is complete in $\mathcal{H}_\psi$. This basis can in principle be constructed from any Hermitian operator $\mathcal{A}$ acting on the Hilbert space $\mathcal{H}_\psi$ using the Gram-Schmidt procedure as in (5). One can expand the target state in the same basis, the coefficients of this expansion being functions of $s$ as

$$|\psi(s)\rangle = \sum_{n=0}^{\mathcal{D}_\psi - 1} \psi_n(s)|n\rangle, \tag{15}$$

which yields, from (14),

$$\mathcal{C}(s) = \sum_{n=0}^{\mathcal{D}_\psi - 1} n|\psi_n(s)|^2. \tag{16}$$

In principle, the coefficients $\psi_n(s)$ can be recursively read off from the equation (12) if one is equipped the full knowledge of Lanczos coefficients $a_n$ and $b_n$.

Let us now denote the eigenstates and eigenvalues of the operator $\tilde{\mathcal{A}}$ by $\{|\alpha_i\rangle\}$[1] and $\alpha_i$, respectively. Since $\tilde{\mathcal{A}}$ is Hermitian, these eigenstates form a complete set of states in $\mathcal{H}_\psi$. Using this fact, the spreading number (14) can be recast into the following form

$$\mathcal{C}(s) = \sum_{\alpha_1,\alpha_2} U(\alpha_1,s)U^*(\alpha_2,s)\rho_0(\alpha_1,\alpha_2)\langle\alpha_1|\ell|\alpha_2\rangle, \tag{17}$$

where $\rho_0(\alpha_1,\alpha_2) = \langle\alpha_1|\psi(0)\rangle\langle\psi(0)|\alpha_2\rangle$ is the density matrix in the $\alpha$-basis. $\langle\alpha_1|\ell|\alpha_2\rangle$ are the matrix elements of the label operator in eigenvectors of the Hermitian operator $\tilde{\mathcal{A}}$ given by

$$\langle\alpha_1|\ell|\alpha_2\rangle = \sum_{n=0}^{\mathcal{D}_\psi} n\langle\alpha_1|n\rangle\langle n|\alpha_2\rangle. \tag{18}$$

It is worth noting here that these matrix elements can be computed directly in the continuum limit using the recursion relation given in (5). Expanding $|n\rangle$ in the basis of eigenvectors of the operator $\tilde{\mathcal{A}}$

$$|n\rangle = \sum_i c_n(\alpha_i)|\alpha_i\rangle \tag{19}$$

the equation (5) reads

$$\alpha c_n(\alpha) = a_n c_n(\alpha) + b_n c_{n-1}(\alpha) + b_{n+1} c_{n+1}(\alpha), \tag{20}$$

where $c_n(\alpha) = \langle\alpha|n\rangle$, $|\alpha\rangle$ being a particular eigenstate of $\tilde{\mathcal{A}}$ with non-degenerate eigenvalue $\alpha$. This equation can be thought of as a time independent Schrödinger equation for which one can find the wave functions $c_n(\alpha)$ recursively. Using these wave functions, one can further compute the matrix elements (18).

In order to obtain an expression for the matrix elements in the continuum limit, one can first rescale $c_n(\alpha) \rightarrow (i)^n c_n(\alpha)$ and then set $x = n\epsilon$, $b(x) = 2\epsilon b_n$, $a(x) = a_n$ and $c_n(\alpha) = c(x,\alpha)$. Thereafter, expanding (20) upto the leading order in $\epsilon$ yields a much simpler equation

$$-i(\alpha - a(y))f(y,\alpha) = \partial_y f(y,\alpha), \tag{21}$$

where $\partial_y = b(x)\partial_x$ and $f(y,\alpha) = \sqrt{b(x)}c(x,\alpha)$. The equation (21) can be readily solved to obtain

$$f(y,\alpha) = f(0,\alpha)e^{-i\alpha y + i\int_0^y a(y')dy'}. \tag{22}$$

Using this solution, finally one arrives at an expression for the matrix element (18) as

$$\langle\alpha_1|\ell|\alpha_2\rangle = \frac{1}{\epsilon^2}\int dy\, x(y)e^{-i\alpha_{12}y}, \tag{23}$$

where $\alpha_{12} = \alpha_1 - \alpha_2$. As its form suggests, to evaluate the matrix elements explicitly, one needs to know $x$ as a function of $y$ which can be obtained from the relation $\frac{dx}{b(x)} = dy$. Given $b(x)$, the integral can be performed to find $x(y)$ and subsequently, the explicit expression for the desired matrix elements which determine the spread in the continuum limit.

---

[1]$\{|\alpha_i\rangle\}$ do not in general form an ordered basis.

**Minimising the spreading and complexity:** Since the flow itself is generated in terms of a particular Hermitian operator $\tilde{\mathcal{A}}$, it is natural to consider a special case when $\tilde{\mathcal{A}} = \mathcal{A}$ that is when ordered basis is constructed using the same Hermitian operator that generates the unitary flow (12). One can show that in this case the spreading number is minimum compared to any other choice of Hermitian operator or equivalently, for any other choice of basis. This follows directly from theorem 1 of [1], by setting $n = 1$.

Therefore the resultant spreading number obtained with the choice of $\tilde{\mathcal{A}} = \mathcal{A}$ can be considered as the complexity of the s-evolved target state. Indeed, looking at equation (14) one finds that the minimum of this with $\{|n\rangle\}$ being the orthonormal basis generated by the Hermitian operator $\mathcal{A}$ is of the same form as that of the Krylov complexity [4, 18–20] (see also [21–23]).

It is worth mentioning that due to the property of the trace, the complexity defined above at given $s$ may also be given by

$$\mathcal{C}(s) = \mathrm{Tr}(\ell(s)\rho), \tag{24}$$

that means, we keep the initial reference state fixed and construct the ordered basis using the Hermitian operator $\mathcal{A}$ appearing in the evolution operator $\mathcal{U}(\mathcal{A}, s)$.

**The time-evolution as an example:** As a particularly interesting and rather physical example of the generalized framework developed above, let us consider the case where the Hermitian operator $\mathcal{A}$ is the Hamiltonian of the quantum system. This is the example mainly studied in the literature to compute Krylov complexity. In this case, starting with an initial state $|\psi(0)\rangle$, the dynamics is given by the Schrödinger equation

$$|\psi(t)\rangle = e^{iHt}|\psi(0)\rangle. \tag{25}$$

Then the density matrix associated with this state at any time $t$ is given by

$$\rho(t) = |\psi(t)\rangle\langle\psi(t)| = e^{iHt}\rho(0)e^{-iHt}, \tag{26}$$

where $\rho(0) = |\psi(0)\rangle\langle\psi(0)|$, by which the complexity is

$$\mathcal{C}(t) = \mathrm{Tr}(\ell\rho(t)) = \sum_{n=0}^{\mathcal{D}_\psi - 1} n\,|\psi_n(t)|^2, \tag{27}$$

where $\psi_n(t)$ is given by the following expression where the state is expanded in terms of the ordered basis $\{|n\rangle\}$ (known as Krylov basis in this case)

$$|\psi(t)\rangle = \sum_{n=0}^{\mathcal{D}_\psi - 1} \psi_n(t)|n\rangle. \tag{28}$$

Using the Schrödinger equation, one can deduce an equation for $\psi_n(t)$ that can be solved recursively using the knowledge of Lanczos coefficients $a_n$ and $b_n$. Actually for a chaotic system these coefficients, as functions of $n$, follow a universal behavior that includes a linear growth [4] followed by a saturation [18–20]. These behaviors give an early-time exponential and the late times linear growths for complexity respectively [18–20].

In order to proceed further to explore the complexity associated with the Hamiltonian evolution, let us assume that the Hamiltonian of the system has a continuous spectrum. Therefore, using the energy eigenstates, the Krylov complexity (27) may be recast into the following form

$$\mathcal{C}(t) = \int dE_a\,dE_b\,e^{i(E_a - E_b)t}\rho_0(E_a, E_b)A(E_a, E_b), \tag{29}$$

where

$$\rho_0(E_a, E_b) = \langle E_a | \rho(0) | E_b \rangle, \qquad A(E_a, E_b) = \langle E_a | \ell | E_b \rangle, \tag{30}$$

which is the same expression proposed for complexity in [14]. In [14], the left hand side of (29) was identified as complexity provided the function $A(E_a, E_b)$ had a double pole structure at late-time, i.e. in the coincident limit $E_a \to E_b$, thereby restricting the function $A(E_a, E_b)$ to a class of quantum expectation values which violates the ETH. We shall argue below that this is naturally the case when we consider the expectation value of the label operator which turns out to be the atypical non-local operator existence whereof was postulated in [14].

As we have already mentioned the saturation of Lanczos results in a linear growth at late times. In the notation of (29) this behavior at late times imposes a condition on the function $A(E_a, E_b)$ to have a double pole structure

$$A(E_a, E_b) = -\frac{a(E)}{\omega^2} + \text{local terms}, \quad \text{for} \quad \omega \to 0, \tag{31}$$

where $\omega = E_a - E_b$ and $2E = E_a + E_b$, in agreement with the proposal of [14]. We can derive the surprising connection between the double pole structure of the function $A(E_a, E_b)$ and the saturation of Lanczos coefficients from the expectation value of the label operator in the continuous limit given in (23). When the Lanczos coefficient $b(x)$ saturates to a constant, one gets $x = y$. This yields, from the expression (23),

$$\langle E_a | \ell | E_b \rangle = \frac{1}{\epsilon^2} \int_0^\Lambda dy \, x(y) e^{-i\omega y} = -\frac{1}{\omega^2} (1 - e^{-i\omega\Lambda}), \tag{32}$$

where $\Lambda$ is a cutoff. This is the double pole structure we expect to get from $A$-function that generates linear growth for complexity at late time. For $\Lambda \to \infty$ the above integral may be recast into the following form

$$\langle E_a | \ell | E_b \rangle = \frac{1}{\epsilon^2} \int_0^\infty dy \, x(y) e^{-i\omega y} = \frac{1}{\epsilon^2} i \frac{d}{d\omega} \delta(\omega). \tag{33}$$

On the other hand, in the regime when the Lanczos coefficient $b(x)$ exhibits a linear growth, say, $b(x) = 2\lambda x$, one obtains the relation $x = e^{2\lambda y}$, which upon inserting in equation (23) yields

$$\langle E_a | \ell | E_b \rangle = \frac{1}{\epsilon^2} \delta(\omega + 2i\lambda). \tag{34}$$

Plugging this result in (29) one finds early time exponential growth $\mathcal{C}(t) \sim e^{2\lambda t}$ as the case for chaotic system.

For a more general case of $x = y^m$ for $m > 1$, one gets

$$\langle E_a | \ell | E_b \rangle = \frac{1}{\epsilon^2} i^m \frac{d^m}{d\omega^m} \delta(\omega), \tag{35}$$

which can happen at early times in a non-chaotic system. Therefore, from (29) one gets $\mathcal{C}(t) \sim t^m$ which can be interpreted as the early time power-law growth for complexity.

An advantage to study the matrix elements of the label operator is that the universal behavior of Lanczos coefficients associated with a Hermitian operator may be studied without refereeing to a particular dynamics by which a state evolves.

To explore the significance of the label operator better, we note that for an arbitrary operator $\Lambda = \sum_a \lambda_a |\lambda_a\rangle\langle\lambda_a|$ where $\lambda_a$ and $|\lambda_a\rangle$ are its eigenvalues and eigenstates, respectively, one can compute $\text{Tr}(\Lambda\rho(t))$ which has the same form as that of (29), though in this case it is not guaranteed that the functions $A(E_a, E_b) = \langle E_a | \Lambda | E_b \rangle$ exhibit double pole structures at

late times. Actually we would expect that for a typical operator $\Lambda$ this satisfies the ETH ansatz and thus $\text{Tr}(\Lambda\rho(t))$ gives a time independent quantity interpreted as the average of the operator $\Lambda$. Therefore in order to have the notion of complexity it is important to compute (29) specifically for the label operator of an ordered basis. This justifies further the uniqueness of our definition of complexity in terms of the label operator.

The other important element of the formula given in (29) is the density matrix $\rho_0(E_a, E_b)$. At the leading order in the dimension of the Hilbert space, the density matrix $\rho_0(E_a, E_b)$ is factorized, though in general it has a form

$$\rho_0(E_a, E_b) = \rho(E_a)\rho(E_b) + \rho_c(E_a, E_b). \tag{36}$$

Here $\rho_c$ represents the connected term meaning that it cannot be written in a factorized form of $g_1(E_1)g_2(E_2)$ with $g_{1,2}$ being arbitrary functions of energy. While this object remains a silent spectator in the discussion of the late time linear growth of complexity above, we will now argue that it plays a pivotal role in understanding saturation phase of complexity at later times.

**On the saturation of complexity at late time:** In what follows we would like to present a general form of complexity in $\tau$ scaling limit in which we take $\{t, \mathcal{D}_\psi\} \to \infty$ while keeping $\tau = t\,\mathcal{D}_\psi^{-1}$ fixed.

We proceed by rescaling $\rho_0(E_a, E_b) = \mathcal{D}_\psi^2\tilde{\rho}_0(E_a, E_b)$ and switching to the $(E, \omega)$ coordinates. The latter is convenient for studying the coincident limit $E_a \to E_b$ relevant for the late time limit $t \to \infty$. With this, the complexity (29), up to an appropriate normalization, reads

$$\mathcal{C}(t) = \int_0^\infty dE \int_{-\infty}^\infty d\omega\, e^{i\omega t}\tilde{\rho}_0(E, \omega)A(E, \omega). \tag{37}$$

As we have already demonstrated above, the saturation phase of the Lanczos coefficients, which corresponds to the $\tau$ scaling limit, results in a double pole structure for $A(E, \omega)$. Assuming to have an expression for complexity growth at leading order consistent with Lloyd's bound one arrives at [14]

$$A(E, \omega) = -\frac{\sqrt{E}}{\tilde{\rho}(E)\,\omega^2} + \text{local terms}. \tag{38}$$

On the other hand, at the $\tau$ scaling limit, the physics is dominated by correlations between nearby energy levels. In this limit, the connected part of the matrix elements of the density matrix denoted by $\rho_c(E_a, E_b)$ in (36), are described by the universal sine-kernel formula [24]. More precisely, one has

$$\tilde{\rho}_0(E, \omega) = \tilde{\rho}(E)^2 + \frac{\tilde{\rho}(E)}{\mathcal{D}_\psi}\delta(\omega) - \frac{\sin^2(\mathcal{D}_\psi\tilde{\rho}(E)\omega)}{(\mathcal{D}_\psi\omega)^2}. \tag{39}$$

Putting everything together, one arrives at

$$\mathcal{C}(t) = C_0 - \mathcal{D}_\psi^{-1}\int_0^\infty dE\sqrt{E}\int_{-\infty}^\infty d\omega\,\frac{e^{i\omega t}}{\omega^2}\delta(\omega)\int_0^\infty dE\sqrt{E}\tilde{\rho}(E)\int_{-\infty}^\infty d\omega\,\frac{e^{i\omega t}}{\omega^2}$$
$$\times\left(1 - \frac{\sin^2(\mathcal{D}_\psi\tilde{\rho}(E)\omega)}{(\mathcal{D}_\psi\tilde{\rho}(E)\omega)^2}\right), \tag{40}$$

where $C_0$ is a constant. It is exactly the same expression which was obtained for (super)JT gravity in which the density $\tilde{\rho}$ is known explicitly [11–13]. In this equation, the first line

which is divergent in general and needs to be regularized by a cutoff, is nevertheless time independent and does not contribute at late times. Therefore, in what follows we only need to consider the last term. Setting $\omega t = \xi$, this term yields

$$\mathcal{C}(t) = -\mathcal{D}_\psi \, \tau \int_0^\infty dE \sqrt{E} \tilde{\rho}(E) \int_{-\infty}^\infty d\xi \, \frac{e^{i\xi}}{\xi^2} \left( 1 - \frac{\sin^2(\frac{\tilde{\rho}(E)}{\tau}\xi)}{(\frac{\tilde{\rho}(E)}{\tau}\xi)^2} \right), \qquad (41)$$

which vanishes for $\tau > \tilde{\rho}(E)$ for any density function $\tilde{\rho}$ [11]. This behaviour is quite universal in the sense that the saturation phase does not depend on the details of the model. On the other hand, for $\tau < \tilde{\rho}(E)$ one can expand the sine function in terms of exponential functions and performing the complex integral over a contour excluding the poles in the lower half plane one obtains [11, 12]

$$\mathcal{C}(t) \approx -\frac{2\pi \mathcal{D}_\psi}{3} \int_{E_\tau}^\infty dE \sqrt{E} \, \tilde{\rho}^2(E) \left( 1 - \frac{\tau}{2\tilde{\rho}(E)} \right)^3, \qquad (42)$$

where $E_\tau$ can be read off from the equation $\tau = \tilde{\rho}(E_\tau)$. Although to perform the integration over $E$ one needs to know the explicit form of the density $\tilde{\rho}$, one can still extract certain universal behavior for the complexity. In particular for $\tau \ll 1$ one gets $\mathcal{C}(t) \sim -\alpha_0 \mathcal{D}_\psi + \alpha_1 t$ with $\alpha_{0,1}$ being order one constants. Although the saturation phase occurs at $\tau \sim 1$ limit is also universal, the actual way the complexity approaches the saturation phase is model dependent and is fixed as soon as the density is fixed.

To conclude, we note that for a chaotic system, the complexity has a universal behavior starting with early times exponential growth and linear growth at late times followed by a saturation. It is worth noting that while the early exponential and the late times linear growths are described by the behavior of the $A(E, \omega)$ function, already at leading order, the saturation phase is the consequence of the contribution of the connected part of the density-density short range correlation known as sine-kernel.

This has to be compared with the numerical computations done *e.g.* in [25], where the saturation of complexity was due to the descent phase of the Lanczos coefficients. It would be interesting to understand and compare these two different approaches by which the complexity reaches the saturation phase. An early saturation of complexity due to the breaking of ETH might actually signal the chaotic nature of the system. Work in this direction is in progress [26] and we hope to report on this soon.

## 3 Complexity: The operator-state correspondence

In this section we will extend the algorithm elaborated in the previous section, to the study of complexity corresponding to growth of an operator. To achieve this, generally, one starts with a reference operator and constructs an ordered basis of operators using a Hermitian operator. Then one looks for the spreading of the desired operator over this ordered basis.

Typically, in order to go through this procedure, one needs to define a proper inner product in the space of operators. However, in what follows, we will take a different approach so that the inner product will arise naturally.

**The doubled Hilbert space:** To compute complexity associated with the growth of an operator we will essentially use the same procedure as that of the state complexity simply by making use of the channel-state map [17] which maps an operator to a state.

Let us consider a quantum system described by the Hamiltonian $H$ with eigenstates and eigenvalues $|E_a\rangle$ and $E_a$ respectively. The corresponding Hilbert space is also denoted by $\mathcal{H}$

with dimension $\mathcal{D}$. In this set up, we will consider a generic operator $\mathcal{O} : \mathcal{H} \to \mathcal{H}$ and a complete basis spanning the Hilbert space, denoted by $\{|i\rangle, i = 1, 2, \cdots \mathcal{D}\}$. The matrix elements of the operator in this basis is $\mathcal{O}_{ij} = \langle i|\mathcal{O}|j\rangle$. Following the channel-state duality, one can then define an associated state, through a linear bijection,[2] in an auxiliary doubled Hilbert space $\mathcal{H}_d = \mathcal{H} \otimes \mathcal{H}$ as follows

$$|\psi_{\mathcal{O}}\rangle = \sum_{i,j=1}^{N} \varrho_{ij} |i\rangle \otimes |j\rangle \equiv \sum_{i,j=1}^{N} \varrho_{ij} |i, j\rangle . \tag{43}$$

Here the density of state $\varrho_{ij}$ is proportional to the matrix elements $\mathcal{O}_{ij}$ such that $\sum_{i,j} |\varrho_{ij}|^2 = 1$. More precisely, one has

$$\varrho_{ij} = \frac{\mathcal{O}_{ij}}{\sqrt{\sum_{i',j'} \mathcal{O}_{i'j'}\mathcal{O}_{i'j'}}} . \tag{44}$$

This map would naturally define an inner product between two operators

$$\mathcal{O}_1 \cdot \mathcal{O}_2 \equiv \langle \psi_{\mathcal{O}_1} | \psi_{\mathcal{O}_2} \rangle . \tag{45}$$

Once we have the state in the doubled Hilbert space, we can simply adopt our previously developed formalism for state complexity, now for a state in the doubled Hilbert space. For a given Hermitian operator $\mathcal{A} : \mathcal{H} \to \mathcal{H}$ one can then readily construct an orthonormal and ordered basis in the doubled Hilbert space with the identification of the first state as $|0, 0\rangle = |\psi_{\mathcal{O}}\rangle$. All other elements are obtained using the algorithm given in (5), though, now in the doubled Hilbert space

$$\widehat{|n+1, n+1\rangle} = (\mathcal{A}_d - a_n)|n, n\rangle - b_n|n-1, n-1\rangle , \tag{46}$$

where $|n, n\rangle = b_n^{-1}|\hat{n}, \hat{n}\rangle$ and

$$a_n = \langle n, n|\mathcal{A}_d|n, n\rangle , \qquad b_n = \sqrt{\langle \hat{n}, \hat{n}|\hat{n}, \hat{n}\rangle} , \tag{47}$$

where $\mathcal{A}_d = \mathcal{A} \otimes \mathcal{A}$ is an operator acting on the doubled Hilbert space $\mathcal{A}_d : \mathcal{H}_d \to \mathcal{H}_d$. The resulting ordered basis defines a subspace of the doubled Hilbert space denoted by $\mathcal{H}_{d,\psi_{\mathcal{O}}}$, $\mathcal{D}_{\psi_{\mathcal{O}}}$ being the dimension of this sub Hilbert space.

For a given operator $\mathcal{O}$, the spreading is now defined as

$$\mathcal{C}_{\mathcal{O}} = \text{Tr}(\ell \rho_{\mathcal{O}}), \tag{48}$$

where $\ell = \sum_{n=0}^{\mathcal{D}_{\psi_{\mathcal{O}}}} n|n, n\rangle\langle n, n|$ and $\rho_{\mathcal{O}} = |\psi_{\mathcal{O}}\rangle\langle\psi_{\mathcal{O}}|$ is the density matrix of the state $|\psi_{\mathcal{O}}\rangle$ associated with the operator $\mathcal{O}$.

Following this definition, it is then a straightforward task to compute complexity of an operator that is obtained from a reference operator $\mathcal{O}$ via a unitary evolution. Let us consider the following unitary transformation for an operator

$$\mathcal{O}(s) = \mathcal{U}(\mathcal{A}, s) \, \mathcal{O} \, \mathcal{U}^{-1}(\mathcal{A}, s), \tag{49}$$

so that

$$\rho_{\mathcal{O}}(s) = \sum_{i,j=1}^{\mathcal{D}} \sum_{i'j'=1}^{\mathcal{D}} \varrho_{ij}(s)\varrho_{i'j'}^*(s) |i, j\rangle\langle j', i'| , \tag{50}$$

where $\varrho_{ij}(s) = \langle i|\mathcal{O}(s)|j\rangle / \sqrt{\sum |\mathcal{O}_{ij}|^2}$.

---

[2]This is known in literature as the Choi-Jamiolkowski isomorphism [27].

From this, using the definition of spreading in (48), one gets

$$\mathcal{C}(s) = \text{Tr}(\ell \rho_{\mathcal{O}}(s)) = \sum_{n=0}^{\mathcal{D}_{\psi_{\mathcal{O}}}-1} n|\mathcal{O}_n(s)|^2 \,, \tag{51}$$

where

$$\mathcal{O}_n(s) = \sum_{i,j}^{\mathcal{D}} \varrho_{ij}(s)\langle n,n|i,j\rangle \,, \tag{52}$$

which implies to have the following expansion for the state in terms of ordered basis in the doubled Hilbert space

$$|\psi_{\mathcal{O}}(s)\rangle = \sum_{n=0}^{\mathcal{D}_{\psi_{\mathcal{O}}}-1} \mathcal{O}_n(s)|n,n\rangle \,. \tag{53}$$

One can also take the basis $\{|i\rangle\}$ to be the eigenstates of the Hermitian operator $\mathcal{A}$. Note that, here we considered the evolution of the operator by the same operator $\mathcal{A}$ used to construct the ordered basis in the doubled Hilbert space. Following our argument in the case of state complexity, this ensures that (51) represents minimum spreading and therefore can be interpreted as the complexity of the target operator $\mathcal{O}_s$.

**The time evolution revisited:**   As before, our main interest lies in the study of growth of an operator following the evolution with the Hamiltonian of the system

$$\mathcal{O}(t) = e^{iHt}\mathcal{O}e^{-iHt} \,. \tag{54}$$

In this case, it is natural to use the energy eigenstates for the channel- state map by which the corresponding reference space is given by (43) with the replacement $|i,j\rangle \to |E_a, E_b\rangle$

$$|\psi_{\mathcal{O}}\rangle = \sum_{a,b}^{\mathcal{D}} \varrho_{ab}|E_a, E_b\rangle \,, \tag{55}$$

where $\varrho_{ab}$ is the normalized matrix elements of the operator in the energy basis. At a given time $t$, the corresponding matrix elements are

$$\varrho_{ab}(t) = e^{i(E_a - E_b)t}\varrho_{ab} \,, \tag{56}$$

so that the associated state in doubled Hilbert space is transformed as follows

$$|\psi_{\mathcal{O}}(t)\rangle = e^{iH_- t}|\psi_{\mathcal{O}}\rangle \,, \tag{57}$$

where $H_- = H \otimes 1 - 1 \otimes H$. In order to compute the complexity through minimizing the spreading, following our earlier discussion, one needs to construct the ordered basis, specifically using the Hermitian operator $\mathcal{A} = H_-$. In this case, the general algorithm mentioned above, yields the operator complexity

$$\mathcal{C}(t) = \text{Tr}(\ell \rho_{\mathcal{O}}(t)) = \sum_{n=0}^{\mathcal{D}_{\psi_{\mathcal{O}}}-1} n|\mathcal{O}_n(t)|^2 \,, \tag{58}$$

where $\mathcal{O}_n(t) = \sum_{a,b} \varrho_{ab}(t)\langle n,n|E_a, E_b\rangle$.

In the case when the Hamiltonian possesses a continuum spectrum, one can write down an expression for complexity similar to (29) in the doubled Hilbert space. This will lead to a

late time saturation of complexity followed by a linear growth. The saturation will be guaranteed through the perturbative contact terms and non-perturbative contributions appearing in the density correlation $\langle E_a, E_{a'}|\rho_{\mathcal{O}}(t))|E_b, E_{b'}\rangle$ in the coincident limits $E_a - E_b \to 0$ and $E_{a'} - E_{b'} \to 0$.

To summarize, we note here that in order to study operator complexity for an operator evolving with the Heisenberg equation, one can equivalently run the algorithm developed to compute the state complexity, but in the doubled Hilbert space with the Hamiltonian $H_-$.

It is worth noting that, in this case, the ordered basis, generated using the particular effective Hamiltonian $H_-$ which governs the time evolution of the state, renders the Lanczos coefficients $a_n$ to be zero. This is of course expected due to the fact that the evolution of an operator here is given by the Liouvillian which amounts to have vanishing $a_n$.

Let us now discuss one interesting subtlety of this construction. For quantum system defined on the doubled Hilbert space $\mathcal{H}_d$ whose dynamics is given by $H_-$, the average energy $H_+ = H \otimes 1 + 1 \otimes H$ is a conserved charge. This follows from the fact that $[H_-, H_+] = 0$. In other words, for an operator whose associated state is defined by (55), one should impose the following condition

$$H_+|\psi_{\mathcal{O}}(t)\rangle = E|\psi_{\mathcal{O}}(t)\rangle, \tag{59}$$

where $E = E_a + E_b$ is the average energy which is kept fixed.

The average energy should remain constant during the state evolution (57) and therefore it does not mix states with different energies. In other words the complexity can be computed for each individual sector with fixed average energy [28].

Precisely due to this particular dynamics of the operator growth, the diagonal part of the matrix elements $\mathcal{O}_{ab}$ does not contribute to the operator growth. In other words, restricting ourselves to an operator whose corresponding state in doubled Hilbert space is given by

$$|\psi_{\mathcal{O}}^{(0)}\rangle = \sum_a \varrho_a |E_a, E_a\rangle, \tag{60}$$

one gets trivial dynamics under a unitary time evolution given by $H_-$. Here the index (0) indicates that this state belongs to a subspace with constant $H_-$ (which could be zero).

Actually this defines rather atypical states whose dynamics are, rather naturally, given by $H_+$,

$$|\psi_{\mathcal{O}}^{(0)}(t)\rangle = e^{iH_+t}|\psi_{\mathcal{O}}^{(0)}\rangle. \tag{61}$$

Note that for $\varrho_a = e^{-\beta E_a/2}/\sqrt{\sum_a e^{-\beta E_a}}$ this state corresponds to a thermofield double state with inverse temperature $\beta$. In general $\varrho_a$ could be a complex function having a phase $\varrho_a = |\varrho_a|e^{i\alpha_a}$ which could be thought of the generalized thermofield double state [29, 30].

This apparently confusing role reversal of $H_{\pm}$, mentioned above in the context of the atypical state, is actually not that surprising. In fact, in the construction of the doubling of the Hilbert space in (55), it is assumed that energy eigenstates of the Hamiltonians participating in the doubling are also eigenstates of the time reversal operators, as is the case of most quantum mechanical system. This is, however, not generically true. Taking this into account, explicitly, (55) can be rewritten as

$$|\psi_{\mathcal{O}}\rangle = \sum_{a,b}^{\mathcal{D}} \varrho_{ab}\, \mathcal{T}|E_a, E_b\rangle, \tag{62}$$

where $\mathcal{T}$ is the time-reversal operator, which being an anti-linear operator, results in the normalized matrix elements

$$\varrho_{ab}(t) = e^{i(E_a + E_b)t}\varrho_{ab}. \tag{63}$$

Then the associated evolution of the state in the doubled Hilbert space is given by

$$|\psi_{\mathcal{O}}(t)\rangle = e^{iH_+ t}|\psi_{\mathcal{O}}\rangle. \tag{64}$$

When $|\psi_{\mathcal{O}}\rangle$ is the TFD state, such time evolutions give rise to phase-shifted thermofield doubled states mentioned above.

Therefore, we note that there are two different ways to arrive at the same generalized TFD states, either as a simple one-sided time evolution (61), preserving the time-reversal symmetry of the basis states, or in terms of the doubled Hilbert space structure using energy eigenstates which are not time-reversal symmetric as we derived in (64).

These two points of view correspond to the boundary and bulk perspectives, respectively, in the context of AdS/CFT. The former approach remains faithful to the time-reversal symmetry of the CFT while creating an enlarged phase space of complexified quantum states with asymptotic charges [31], and the latter creates a complexified bulk states through evolution with $H_+$ identified as the bulk Hamiltonian. Accordingly, the additional phase appearing in the generalized TFD state assumes interpretations either due to boundary charges or as topological quantum phases arising as a consequence of bulk non-locality.

This non-locality is quantified through the symplectic structure in the bulk, which although can have local interpretation of time evolution with $H_+$, are globally non-exact, thereby giving rise to the additional phases appearing in the generalized TFD states [32, 33].

**A little more on doubling and the holographic interpretation:** Our construction of doubled basis follows a secret rendition of the Reeh-Schlieder-Theorem. By construction, the domain $\Gamma_\psi$ of any initial given state $|\psi\rangle$ is dense in the full Hilbert space of the theory once we are successfully able to construct the ordered basis using an algebra of operators $\Omega$ that minimize the spreading. Therefore, this state can be dubbed as a cyclic vector. Furthermore, to have a non-vanishing complexity, one needs to start with a state which is not an eigenstate of the operator, which means there cannot be an annihilation operator in the small algebra of operators, $\Omega$. This makes the initial state $|\psi\rangle$ a separating vector.

With these two defining conditions of the Reeh-Schlieder Theorem [34], one can visualize the full algebra of operators acting on the Hilbert space $\Gamma_\psi$ as an entangled algebra $\Omega_d = \Omega \otimes \bar{\Omega}$ where $\bar{\Omega}$ is the commutant algebra which can be obtained using modular automorphism via the Tomita-Takesaki theorem [34]. Furthermore, given the emerging structure of entanglement, it is natural to invoke a doubled basis as in (43) to expand the state and consequently, a doubled ordered basis as in (46) to analyse the growth of the state.

This justifies why the doubling, introduced in (43) using the channel-state map, is essential to cast the operator complexity as a corresponding state complexity.[3]

In the context of holography, such constructions were instrumental in understanding the state-dependent reconstruction of black hole interior [36].[4] We note here, following our unified interpretation of the state and the operator complexities, that the same construction is also pivotal to connect to the the notion of holographic complexity which measures the growing volume of the interior of a black hole.

It is worth mentioning that in the holographic context, there can actually be an infinitely many quantities which possess the same late time behaviour in terms of a linear growth followed by a saturation and in principle, all of them can be dubbed as complexities [14, 37].

---

[3]The realisation we achieve through this unification is quite in line with the Gelfand-Naimark-Segal (GNS) construction discussed in the context of studying operator growths in large $\mathcal{N}$ theories [35].

[4]However, in this case the small algebras were only approximate algebra at large $\mathcal{N}$ with edge effects which were important to realize the existence of the black hole interior. In the language of Lanczos, it would corresponding to only approximate breaking of the algorithm when $b_n$ becomes smaller to a given hierarchy scale, as $\mathcal{N}$ is, in the case of holographic CFT's.

Identifying the *A*-functions in terms of the matrix elements of the label operator potentially removes this ambiguity. Therefore, the interpretation of both, in terms of operator growth, should follow from the same doubling algorithm.

An alert reader might wonder about the identification of the matrix elements of the label operator with the expectation values of a non-local holographic position operator. However, we would like to emphasize that the entanglement between two spacelike separated regions can have an equivalent description in terms of entangling algebras of operators defined on the Hilbert space. While the previous description is more geometric and visually pleasing, the latter one provides a more general notion of entanglement independent of spacetime. Following the interpretation of the doubled Hilbert space in terms of the modular automorphism mentioned above, this generalized notion of entanglement justifies the matrix elements of the label operator being identified with the holographic non-local operator. More details regarding this identification will be elaborated in [26].

## 4 Subregion Krylov complexity

Entanglement entropy or other measures of entanglements are given in terms of the reduced density matrix. This is unlike the complexity the usually is defined for a pure state or an operator in entire space. Nevertheless, subregion complexity and complexities for mixed states have also been studied in the context of holographic complexity [38–41]. The circuit complexity for mixed states in open systems has been also studied in [42–47].

Since the approach used in the literature to study complexity, mainly relied on the state, its generalizations to mixed states or to subregions are not straightforward. On the other hand, in this paper, our construction for complexity (of pure state) is given in terms of a particular trace over the density matrix. An advantage to define complexity in terms of density matrix is that it may be extended for the cases where we are dealing with subregion or mixed states. In these cases one would expect that the definition would be the same and we just need to use reduced density matrix.

Let us consider a quantum system whose Hilbert space can be decomposed into two parts $\mathcal{H} = \mathcal{H}_A \otimes \mathcal{H}_B$. The dynamics of the system is given by a Hamiltonian which, in general, may not be decomposed into two parts acting on $\mathcal{H}_A$ and $\mathcal{H}_B$ separately. Therefore, even if we start with a reference state which is separable, as times goes the state spreading makes it very complicated.

Let us start with a reference state $|\psi\rangle$ and construct the orthonormal, ordered basis using the Hamiltonian of the system. Then it is rather straightforward to compute the reduced label operator by taking trace over subsystem $B$ $\ell_A = \mathrm{Tr}_B(\ell)$. One may also compute reduced density matrix at given time $\rho_A(t) = \mathrm{Tr}_B(\rho(t))$. Then the subregion Krylov complexity is defined by

$$\mathcal{C}_A(t) = \mathrm{Tr}_A(\ell_A \rho_A(t)). \tag{65}$$

It is important to note that although the time evolution of the density matrix is simple and follows from the Schrödinger equation

$$\rho(t) = e^{iHt} \rho(0) e^{-iHt}, \tag{66}$$

for the case of reduced density matrix it is rather involved even for an initial density matrix which is factorized $\rho(0) = \rho_A(0) \otimes \rho_B(0)$ [48]. More precisely, using the fact that the density matrix is positive and normalized, one may write $\rho_B(0) = \sum_\mu \lambda_\mu |\mu\rangle\langle\mu|$ which yields [48]

$$\rho_A(t) = \sum_{\mu,\nu} K_{\mu\nu}(t) \rho_A(0) K_{\mu\nu}^\dagger(t), \tag{67}$$

where $K_{\mu\nu}(t)$ known as the Kraus operators are

$$K_{\mu\nu}(t) = \sqrt{\lambda_\nu} \langle \mu | e^{iHt} | \nu \rangle. \tag{68}$$

In general it is not an easy task to compute complexity for reduced density matrix, though for a special cases where the dynamics of the two subsystem is separable, one can expect to make some progress.

More generally, motivated by the operator-state mapping of the previous section, let us consider the following time dependent density matrix

$$\rho(t) = \sum_{a,b,a',b'} \varrho_{ab}(t)\varrho^*_{a'b'}(t)|E_a, E_b\rangle\langle E_{b'}, E_{a'}|, \tag{69}$$

whose initial density is found by setting $t = 0$. Then the reduced density matrix at given time reads

$$\rho_r(t) = \sum_{a,a',c} \varrho_{ac}(t)\varrho^*_{a'c}(t)|E_a\rangle\langle E_{a'}|, \tag{70}$$

by which the complexity is

$$\mathcal{C}(t) = \mathrm{Tr}(\ell_r \rho_r(t)) = \sum_{a,a',c} \varrho_{ac}(t)\varrho^*_{a'c}(t)A(E_a, E_{a'}). \tag{71}$$

For the case of $\varrho_{ab}(t) = e^{i(E_a \pm E_b)t}\varrho_{ab}$ one finds

$$\mathcal{C}(t) = \sum_{a,a'} e^{i(E_a - E_{a'})t}\rho(E_a, E_{a'})A(E_a, E_{a'}), \tag{72}$$

which has essentially the same form as that we have found for pure state complexity. Here $\rho(E_a, E_{a,}) = \sum_c \varrho_{ac}\varrho^*_{a'c}$. Note that in this expression, the functions $A(E_a, E_{a'})$ are matrix elements of the reduced label operator.

For the maximally entangled case, given by the state

$$|\psi\rangle = \sum_a \varrho_a |E_a, E_a\rangle, \tag{73}$$

$\varrho_a$ being a function of $E_a$ with a possible phase, one can assume to have a separable dynamics as follows

$$|\psi(t)\rangle = f_1(H_1, t)f_2(H_2, t)|\psi\rangle, \tag{74}$$

where $f_i$'s are unitary transformations representing the time evolutions of each subsystem. Then the reduced density matrix turns out to be independent of time and is given by

$$\rho_1(t) = \mathrm{Tr}_2(\rho(t)) = \sum_a |\varrho_a|^2 |E_a\rangle\langle E_a|. \tag{75}$$

Clearly, even though the subsystem specified by the reduced density matrix are complex, the complexity remains constant

$$\mathcal{C}(t) = \sum_a \rho(E_a)A(E_a, E_a), \tag{76}$$

where $\rho(E_a) = |\varrho_a|^2$. Therefore one may conclude that having non- zero entanglement entropy is not enough to crate complexity under unitary separable time evolutions and a direct interaction is needed. In other words to get complexity growth for subsystem the dynamics of entanglement is matter (see (67)).

This feature of complexity demonstrated above hints at the interesting fact that in order to have a growth of complexity, it is not sufficient to be content with the factorized structure of

the density matrix, but rather it needs some interaction. This can alternatively be attributed to a fundamentally non-factorized structure of the Hilbert space as well. In the context of the entanglement structure of a generic quantum system, this connection was discussed in [32,33].

This underlines the precise realization of the ER = EPR paradigm [49]. In the context of complexity, one can expect a very similar conclusion following the discussion above. This opens up the possibility of describing the growth of complexity in terms of connected saddles like replica wormholes which in turn could in principle provide the precise geometric interpretation of the modified replica trick for complexity advocated in [12,13]. Work in this direction is in progress and we hope to report on this soon.

As a final comment, we note that the holographic subregion complexity for a time-dependent geometry, given by the Vaidya metric, has been computed in [50–52]. In the context of gauge-gravity duality, the Vaidya metric provides a holographic description for a thermal quench. It was then possible to explore the time-dependence of holographic subregion complexity in this context. It was shown that the holographic subregion complexity exhibits linear growth up to a maximum value after which it shows a decreasing phase and finally saturates to a constant.

Intuitively, one can see that the Krylov subregion complexity should exhibit similar features. In fact, from equation (72) one can note that the linear growth can indeed be understood from the pole structure of the $A$-function, much similar to what we studied in the context of Krylov complexity. On the other hand, since the information about the Lanczos coefficients is encoded in the $A$-function, the saturation phase can be understood from the fact that the Lanczos coefficients vanish at late times.

It would, of course, be interesting to study Krylov subregion complexity more rigorously and compare it with that of holographic one explicitly.

# 5 Conclusions

To summarize our progress in this paper, we developed a unified formalism to study the state and the operator complexities. The key observation towards the unification is to realize that, within the realm of quantum mechanics, one can think of a channel-state correspondence which helps us to cast the the problem of studying operator complexity, to a corresponding state complexity obtained through the evolution of the dual state living on a doubled Hilbert space.

While this connection itself is quite interesting and thought-provoking, it reveals, on its way to development, a couple of more surprises. First, it connects very naturally to the notion of complexity discussed in the context of holography following two routes - i) realizing the doubling of the Hilbert space for a more generic quantum system in terms of the entanglement structure arising from the Reeh-Schlieder theorem in an axiomatic quantum field theory, thus connecting this procedure to the state-dependent bulk reconstructions [36]. This shows that the state-channel map which connects the state and operator complexities through a doubling of the Hilbert space, does secretly exploit the entanglement algebra imposed through the Lanczos algorithm. This hidden structure of entanglement underlying the Krylov construction, to our knowledge, had not been spelt out in the literature; ii) through a direct identification of the matrix elements of the label operator in the energy basis defined to quantify the spreading in a generic quantum system to the expectation values of the holographic position operator appearing in the quenched geodesic length as in [12–14]. This provides a support for a general definition of complexity in terms of the pole structure of the $A$-function as advocated in [14]. Our construction establishes that this definition of complexity is universally applicable for any quantum system, with or without gravity, so long as the complexity has a late time linear

growth. Such behaviour of complexity is indeed expected for a large class of physical systems, both integrable and chaotic and unlike the saturation phase, it is solely governed by the saturation of the Lanczos coefficients, captured by the pole structure of the *A*-function.

While these two routes lead to a further unification, now also incorporating the holographic complexity along with the state and the operator complexities, there is also a second but very important practical advancement in terms of developing a formalism to study subregion complexity and complexity for mixed quantum states, which had so far been quite illusive in the existing literature. In our formalism, this appears, rather naturally, since our universal definition of complexity is given in terms of a particular trace over the density matrix. In particular, the subregion complexity can be obtained simply by replacing the density matrix with the reduced density matrix corresponding to any given subregion. Furthermore, while generalizing the notion of complexity for a mixed state, we realise that this naturally hints at having a replica wormhole saddles in the dual gravitational spacetime, in line with the expectation coming from the modified replica trick proposed in [12, 13].

Following this short summary let us now conclude with a few ongoing progresses.

**The mutual complexity:**    Since we are dealing with subregion complexity, one can naturally define the Krylov mutual complexity for two subregions $A$ and $B$ as follows

$$\mathcal{M}_{AB} = \mathcal{C}_A + \mathcal{C}_B - \mathcal{C}_{A \cup B}, \tag{77}$$

where $\mathcal{C}$'s are the Krylov subregion complexities associated with the mentioned regions. In the context of the holographic complexity, the mutual complexity has been defined in [41, 53].

**Complexity in open quantum systems:**    Our universal formalism for state and operator complexities is applicable as well for open quantum systems which lack a Hermitian Hamiltonian which in turn means that the ordered basis defined in (5) does not span the full doubled Hilbert space $\mathcal{H}_\psi$. In this case, one needs two independent sets of basis vectors, $|n\rangle$ and $\langle \bar{n}|$ which are not in general orthogonal to each other. Nevertheless, given a non-Hermitian operator $\mathcal{A}$, in many cases,[5] it is possible to construct an anti-linear operator $\mathcal{L}$ such that $\mathcal{L}^{-1} \mathcal{A} \mathcal{L}$ is Hermitian with respect to the inner product $\langle n|\bar{n}\rangle_{\mathcal{L}} \equiv \langle n|\mathcal{L}\bar{n}\rangle$. One example of such an anti-linear operator is the CPT operator. Adopting this formalism for computing complexity due to a non-Hermitian operator $\mathcal{A}$ amounts to a generalization of the definition of complexity (10) as

$$\mathcal{C}_\phi = \mathrm{Tr}(\ell \rho_\phi)_{\mathcal{L}}, \tag{78}$$

for any state $|\phi\rangle \in \mathcal{H}_\psi$. Here the subscript $\mathcal{L}$ reminds us that the trace operation here should be performed taking into account the modified inner product $\langle n|\bar{n}\rangle_{\mathcal{L}}$.

**The transition matrix**    As an aside comment, let us mention that there is an other operator, beside the density matrix, which plays an important role in this context that is known as the transition matrix

$$\tau = |\psi\rangle\langle\phi|. \tag{79}$$

It is natural to consider the case where the state $|\phi\rangle$ is given by (12). Although the information of Lanczos coefficients is encoded in the trace of the transition matrix $\mathrm{Tr}(\tau) = \langle\psi|U^\dagger(\mathcal{A}, s)|\psi\rangle = \psi_0^*(s)$, its evolution does not lead to any spreading as defined by (14). More precisely one has

$$\mathrm{Tr}(\ell \tau) = 0. \tag{80}$$

---

[5]Although this formalism was used and tested for pseudo-Hermitian Hamiltonians with real eigenvalues [54, 55], we expect this procedure to work for more general non-Hermitian operators.

It is important to note that the vanishing of the above equation is a direct consequence of our definition of the label operator in which we set $c_n = n$ that is zero for $n = 0$. If one assumes a general $c_n$ for $c_0 \neq 0$ one has $\text{Tr}(\ell\tau) = c_0\text{Tr}(\tau)$.

Transition matrix is used to define the pseudo-entropy which is the generalization of entanglement entropy with post-selection [56, 57]. Taking into account that the fact that the autocorrelation can be re-expressed in terms of the the Lanczos coefficients, one would expect that the pseudo-entropy exhibits certain universal behavior, at least for chaotic systems. We hope to report on this soon.

## Acknowledgments

We would like to thank Rathindra Nath Das, Moritz Dorband and Johanna Erdmenger for several useful and inspiring discussions.

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
