# Peer review of "A universal approach to Krylov State and Operator complexities"

_SciPost Physics, doi:SciPost Phys. 15, 080 (2023)_

## Round 2 · Referee Report · Anonymous (Referee 1) · 2023-5-8

Strengths

The paper discusses a timely topic of Krylov complexity, attempting to provide a uniform description of Krylov operator and state complexities.

Weaknesses

The formal results presented in the pape are correct, but it is not clear how useful these results are to characterize complexity of real physical systems. Some of the results are based on an assumption of certain behavior, e.g. as in eq. (31) , while it is not clear if the operator complexity for any reasonable physical system will behave this way.

Report

Krylov complexity, and dynamics in Krylov space in general, have drawn significant attention recently. There are different proposals how dynamics in Krylov space can be used to probe chaos, and complexity growth of quantum systems. The original work of Parker et al. discussed operator growth, but more recently Krylov formalism was applied to quantum states to characterize complexity of the latter. This paper proposes a uniform approach, but representing operators as states in some extended Hilbert space. This idea is technically correct, but the paper is less clear about practical or conceptual benefits of such an approach. It is very clear that the dynamics in Krylov space is very different for operators and states (say in the former case Lanczos coefficients often grow linearly, while in the latter case their growth saturates to a constant), and thus similarity between the two is only formal.
  • validity: ok
  • significance: ok
  • originality: good
  • clarity: ok
  • formatting: good
  • grammar: good

Author:  Souvik Banerjee  on 2023-05-29  [id 3693]

(in reply to Report 1 on 2023-05-08)
Category:
answer to question
reply to objection
pointer to related literature

We would like to thank the referee for going through our manuscript in detail and for their valuable comments. We would like to respond to the referee's comments as follows. In particular, we will address categorically the following three main concerns raised by the referee, namely,

  1. whether the late time double pole structure of the $A$-function is valid for any "reasonable quantum system'',
  2. practical benefits of the unified notion of Krylov state and operator complexities, and
  3. conceptual benefits of the unification.

Our response on the double pole structure: We know that for a large class of systems, at late times, complexity is expected to grow linearly with time. Examples include both integrable and chaotic systems. Given the fact that there are two contributions in complexity, namely the spectral contribution and the Lanczos dynamics, it is clear that this late-time linear growth should be attributed to the late-time dynamics of Lanczos coefficients. Equation (29) of the draft is the expression of complexity in the energy basis, where the A-function captures the aforementioned dynamics of Lanczos coefficients. Indeed as argued in reference [14], a late-time linear growth of complexity singles out the double pole structure for the $A$-function such that the label operator is not a standard local operator satisfying ETH. We also demonstrated the same explicitly in and following equation (32). Therefore, the pole structure as in equation (31) of the draft is only a consequence of late-time linear growth of complexity. The main statement in this context is, so long the complexity has a late time linear growth, the corresponding $A$-function in the energy basis should exhibit the double pole structure. Immediate examples of reasonable quantum systems where such behaviours were studied are the JT gravity models. These models essentially describe non-relativistic quantum particles in Morse-like potentials (see ref [12]). Both in the supersymmetric (see ref [13]) and non-supersymmetric (see ref [12], ref [11]) cases, one can derive the late time linear growths of complexity. Moreover, in a very recent work https://arxiv.org/abs/2303.12151, it was argued that this linear behaviour of complexity is guaranteed for any quantum system through a generalized Ehrenfest theorem in Krylov space and is therefore not a diagnostic of quantum chaos. Hence, these can all be thought of as nice examples where one can expect the A-function to have exactly the same pole structure as advocated in equation (31) of our draft. Here we would like to clear up one possible source of confusion. In comparison to the time scale of saturation of complexity in the chaotic systems which is $t\sim e^{S_0}$, sometimes the linear growth phase regime is referred to as an "early time regime" in literature. Our claim is that the behaviour of complexity in this regime is universally captured by the pole structure of $A$-function as in equation (31) of our draft.

Our response on practical benefits: The main practical benefit of our uniform approach to complexity lies in the general expression for complexity given as a trace over the product of density function and label operator, as in equation (14) or, equivalently, equation (24). With the state-channel mapping, this universal expression for complexity also encompasses the operator complexity as shown through equation (51) of our draft. While, naively, this looks like a formal expression, the practical advantage is multifold. First, since the universal expression is given in terms of a particular trace over the density matrix, ** it is very easy to generalize this to the notion of subregion complexity** simply by replacing the density matrix with the reduced density matrix corresponding to any given subregion. We elaborated on this very important consequence of our formalism in section IV of our draft. This is an elegant way to understand subregion complexity, which has so far been elusive in the existing literature. Second, the general definition of complexity in terms of trace becomes pivotal to compute quantities like mutual complexity, and even complexity for a general open quantum system. While we have small discussions on these applications in the present conclusion section of our draft, works in these directions are in progress.

Our response on conceptual progress We discussed how the universal approach to complexity offers practical benefits to compute several information quantities including subregion complexity, which can of course also be considered as a novel conceptual development on its own. Actually, our approach to unifying state and operator complexities offers more fundamental conceptual advancements. In particular, our approach reveals the hidden structure of entanglement underlying the Krylov construction. To our knowledge, this connection, which we elaborate on in section IIIc, had not been spelt out in the literature. We showed that the state-channel map which connects the state and operator complexities through a doubling of the Hilbert space, does secretly exploit the entanglement algebra imposed through the Lanczos algorithm. Last but not least, the universal formalism we advocated in this paper, also provides with a fundamental understanding of the AdS/CFT duality in the light of operator complexity and the corresponding description of state complexity via time evolution. In section IIIb, we discussed how the choice of energy basis, being time-reversal symmetric or not, provides two different descriptions of the complexification of TFD states, which can be identified as the boundary and the bulk descriptions in the AdS/CFT correspondence respectively. While in the first case, the complexification is understood in terms of adding asymptotic charges, in the latter description this is due to the topology of the bulk spacetime yielding manifest non-locality. As we briefly discussed following equation (64), our universal approach to complexity provides a fundamentally new viewpoint towards understanding how the manifestly non-local dynamics in the bulk are captured by an effective local boundary theory.

We hope that the referee is now satisfied with our explanations.

Yours sincerely, Mohsen Alishahiha and Souvik Banerjee

---

## Round 2 · Referee Report · Anonymous (Referee 2) · 2023-6-13

Strengths

1.The paper provides a unified formalism to address operator and state Kylov complexities. 2. The paper is well-written, and the relevant equations are presented in sufficient detail. 3. The content is of interest to the quantum gravity and AdS/CFT community, and may also be of interest to people working on quantum chaos and many-body dynamics.

Weaknesses

  1. The discussion is rather formal, with very few explicit computations.
  2. The article advocates an approach to defining subregion complexity, but does not discuss much the properties of this definition, or applications to AdS/CFT.

Report

The article studies Krylov complexity in general quantum systems. A unified framework for the discussion of state and operator Krylov complexities is discussed by using the map from operators to states in a doubled-Hilbert space. This language is then also used to advocate a definition of subregion Krylov complexity. Overall, the article is well-written and all the details are presented clearly. The content is of broad interest to the quantum gravity and AdS/CFT community, and more generally also to people working on quantum chaos and many-body dynamics.

I found the discussion rather formal. In my opinion, the paper could have used more explicit calculations (for e.g., in some representative models of integrable/chaotic dynamics) which would give more confidence in the applicabiity of the formal ideas presented in the article. In the same spirit, it is not clear to me whether the definition of subregion complexity advocated in the article is useful in that it has some nice properties, or whether it has any applications to holography.

Nevertheless, overall I think the article makes a useful contribution to the current and active field of quantum complexity in many-body systems. In my opinion, the article merits publication.

---

## Round 3 · Author Response

List of changes
1. We have added a couple of paragraphs at the end of section IV on the possible connection between holographic subregion complexity and Krylov subregion complexity.
2. We have modified and extended the conclusion section focussing on the practical usefulness and conceptual novelty of our work.

---

## Round 3 · List of Changes

1. We have added a couple of paragraphs at the end of section IV on the possible connection between holographic subregion complexity and Krylov subregion complexity.
2. We have modified and extended the conclusion section focussing on the practical usefulness and conceptual novelty of our work.

---

## Editorial Decision

published